# A Direct Position Determination Method Based on Subspace Orthogonality in Cross-Spectra under Multipath Environments

**DOI:** 10.3390/s22197245

**Published:** 2022-09-24

**Authors:** Kehui Zhu, Hang Jiang, Yuchong Huo, Qin Yu, Jianfeng Li

**Affiliations:** 1College of Electronic Information Engineering, Nanjing University of Aeronautics and Astronautics, Nanjing 211106, China; 2Hai Hua Electronic Enterprise (China) Corporation, Guangzhou 510670, China

**Keywords:** direct position determination, subspace orthogonality, cross-spectra, multipath environment, distributed sensors

## Abstract

Without the estimation of the intermediate parameters, the direct position determination (DPD) method can achieve higher localization accuracy than conventional two-step methods. However, multipath environments are still a key problem, and complex high-dimensional matrix operations are required in most DPD methods. In this paper, a time-difference-of-arrival-based (TDOA-based) DPD method is proposed based on the subspace orthogonality in the cross-spectra between the different sensors. Firstly, the cross-spectrum between the segmented received signal and reference signal is calculated and eigenvalue decomposition is performed to obtain the subspaces. Then, the cost functions are constructed by using the orthogonality of subspace. Finally, the location of the radiation source is obtained by searching the superposition of these cost functions in the target area. Compared with other DPD methods, our proposed DPD method leads to better localization accuracy with less complexity. The superiority of this method is verified by both simulated and real measured data when compared to other TDOA and DPD algorithms.

## 1. Introduction

Passive location technology has increasingly become an important location method because of its strong concealment, and it has been widely studied and applied in both military and civil fields [1]. The positioning stations can be classified into array-based and single-sensor-based, and single-sensor-based stations are lower-cost. As a significant composition of passive location, the two-step location method first requires a measurement of intermediate parameters, such as direction of arrival (DOA) [2,3,4,5], time of arrival (TOA) [6,7], time difference of arrival (TDOA) [8,9,10], or received signal strength (RSS) [11], which are then used to establish the positioning equation and calculate the target position.

In a multi-station positioning scenario, the TDOA-based method is famous for its low complexity and availability in real time, the key technology of which is time delay estimation (TDE). The most classical TDE method is based on generalized cross-correlation (GCC), in which the similarity of two signals is compared by cross-correlation technology [12]. In [13], a new method for effectively estimating maximum likelihood frequency weighting is proposed in the framework of GCC time delay estimation. Another typical TDE method is based on the construction of a cost function with optimization criteria such as maximum likelihood (ML) and minimum mean square error (MMSE) [14,15], using searching or iterative operations to obtain the TDE. In the practical application scenario, due to the complexity and diversity of channel propagation and the environment around the receiver, there tends to be multipath propagation between the source signal and the observation stations [16]. In [17], a TDE method based on ML is proposed for application in multipath environments. In order to overcome the influence of multipath enviroments, a kind of super-resolution TDE method is proposed on the basis of subspace decomposition [18,19] inspired by multiple signal classification (MUSIC) [20,21]. In [22], a field programmable gate array (FPGA) implementation is introduced based on TDOA, which is suitable for multipath environments. In recent years, deep learning (DL) has been applied to estimate the emitter location, especially in the indoor positioning scenario. In [23], a convolutional neural network (CNN) is trained with raw channel impulse response (CIR) and ground truth positional data, and this CNN is proven to be useful in position estimation for multipath effects. In [24], a robust TOA estimation method is proposed based on CNN on randomized channel models, which outperforms the classic positioning methods in low-SNR situation.

In two-step methods, the measurement process of intermediate parameters is independent of the calculation to the target position, which leads to the lack of spatial geometric constraints between the target and observation stations. Different from two-step methods, the one-step direct position determination (DPD) method does not need to estimate the intermediate parameters and is proven superior to traditional two-step methods in accuracy for low signal-to-noise ratios (SNRs) [25,26]. In [27], the performance of DPD is discussed by Amar and Weiss when model errors exist, and it is concluded that the DPD method performs better than conventional DOA methods in multipath propagation situations. The authors in [28] propose a DPD method based on a single moving coprime array by applying subspace data fusion (SDF), which outperforms traditional two-step methods. In [29], a multi-array data fusion based (MDF) DPD method is proposed by using quantum-behaved particle swarm optimization (QPSO) to search for the best array response. In [30], a high resolution DPD method of multiple emitters based on minimum-variance-distortionless-response (MVDR) is proposed, which does not require the prior knowledge of the number of signals. In [31], the DPD methods in the light of coherent and noncoherent pre-processing techniques of wideband signals are proposed, increasing the location accuracy of DPD. In [32], the DPD estimator of indoor radio sources for hybrid antennae is derived, which has better robustness to multipath interference in indoor environments. The DPD methods are originally presented on the basis of arrays, which tend to have high hardware cost.

In order to avoid the expensive hardware cost caused by array-based DPD methods, TDOA-based DPD methods emerge, in which each observation station is equipped with single sensor. In [33], the authors construct an efficient determinant-based cost function with an orthogonal relationship between the received signals and noise in distributed sensor scenarios. TDOA-based DPD method can be realized by the maximum likelihood estimation (MLE) cost function [34,35], which requires a lower amount of calculation. In [36], a maximum correlation cumulative DPD method is proposed for four-station TDOA location scenarios based on electronic elevation map search. In [37], a DPD method for multiple emitters transmitting unknown linear frequency modulation (LFM) signals is proposed based on short time Fourier transform (STFT) and Hough transform (HT). In practice, clock biases may appear in TDOA location scenarios, which may negatively affect the positioning accuracy. In [38], the authors calibrate clock biases jointly by exploiting anchor sources, and perform coarse and refined parameter estimation with an expectation-maximization (EM) algorithm and a Gauss–Newton algorithm respectively. In [39], the authors proposed a DL-based direct position estimation to the mobile objects by the CIR extracted at the receivers, which works well even under harsh multipath propagation. Though TDOA-based DPD methods outperform the conventional two-step methods in accuracy, their complexity is still higher than TDOA methods because of the high dimensional matrix operation. Moreover, in most of the TDOA-based DPD methods, performance in multipath environments is not presented, which may make them hard to apply in practice.

In this paper, a TDOA-based DPD method in multipath environments is proposed based on the subspace orthogonality in cross-spectra between received signals and reference signal. The received long data are first segmented to reduce the complexity of subsequent data processing. The cross-spectrum is obtained by performing Fourier transform (FT) on cross-correlation between received and reference data. The effective part with fixed length is selected in each cross-spectrum, and its covariance matrix is decomposed into signal and noise subspace. The cost function of each spectrum is built based on the subspace orthogonality, and the target position can be obtained by searching the superposition of cost functions in the interested area. In this paper, the low complexity is realized by data segmentation and screening of the effective parts of the data, and the data coherent is solved by the process of spatial smoothing. The proposed method is compared with the TDOA method based on GCC in [12], the TDOA method based on the normalized cross-spectrum (NCS) in [18], and the determinant-based DPD method in [33] with simulated data. The effectiveness and superiority of the proposed method are verified by simulation and real-world test results.

## 2. Signal Model

The TDOA-based location scenario consisting of several single-sensor stations in a multipath environment is shown in Figure 1. The number of sensors is L(L≥3), the positions of which are denoted by $q_{1},q_{2},\ldots ,q_{L}$, and the source location is denoted by p. Set $q_{1}$ as the reference sensor. After receiving the signals from the target source, the sensors send them to the central station for unified processing. The discrete received signal in the multipath environment of the sensors can be expressed by
(1)$$\left\{\begin{array}{cc} x_{1}\left(n\right)=\sum_{m=1}^{M}\lambda_{1m}s\left(n-\tau_{1m}\right)+\sigma_{1}\left(n\right) & \\ x_{l}\left(n\right)=\sum_{m=1}^{M}\lambda_{lm}s\left(n-\tau_{lm}\right)+\sigma_{l}\left(n\right) \end{array}\right.l=2,3,\ldots ,L,n=1,2,\ldots ,N_{l}$$
where *M* represents the number of multipath components, which is obtained by the Akaike information criterion (AIC) or minimum description length (MDL) criterion and is considered as a known quantity to simplify the subsequent deduction, $N_{l}$ is the length of the received signal, $\lambda_{1m}$ and $\lambda_{lm}\left(m=1,2,\ldots ,M\right)$ are the amplitude coefficients in multipath environment, s(n) is the unknown transmit signal, $\sigma_{1}\left(n\right)$ and $\sigma_{l}\left(n\right)$ are zero-mean additive Gaussian noise, and $\tau_{1m}$ and $\tau_{lm}$ are the time delay between the *m*-th multipath component in the corresponding received signal and transmit signal.

Assume that there is no multipath component in the reference signal for the convenience of derivation. In fact, the signal with prior information is chosen to be the reference signal. We have $\lambda_{1m}=\lambda_{1},\tau_{1m}=\tau_{1}$. Then, Equation (Equation 1) can be rewritten as
(2)$$\left\{\begin{array}{cc} x_{1}\left(n\right)=s_{1}\left(n\right)+\sigma_{1}\left(n\right) & \\ x_{l}\left(n\right)=\sum_{m=1}^{M}\mu_{lm}s_{1}\left(n-\Delta \tau_{lm}\right)+\sigma_{l}\left(n\right) \end{array}\right.l=2,3,\ldots ,L,n=1,2,\ldots ,N_{l}$$
where $s_{1}\left(n\right)=\lambda_{1}s\left(n-\tau_{1}\right)$, $\mu_{lm}=\lambda_{lm}/\lambda_{1}$, $\Delta \tau_{lm}=\tau_{lm}-\tau_{1}$ is the time delay difference between other multipath signal and the reference signal, $\tau_{1}=∥q_{1}-p∥/c$; and *c* represents light’s velocity.

## 3. Proposed Scheme

### 3.1. Data Pre-Processing

In order to reduce the amount of subsequent matrix operations, the received data with length $N_{l}$ are evenly segmented into *K* parts, each part with length $N_{0}=N_{l}/K$. The premise of data segmentation to each received signal is that it does not change the correlation between the received signals. Define the *k*-th part of $x_{1}\left(n\right)$ and $x_{l}\left(n\right)$ as $x_{1k}\left(n_{k}\right)$ and $x_{lk}\left(n_{k}\right),k=1,2,\ldots ,K$, $n_{k}=1+\left(k-1\right)N_{0},2+\left(k-1\right)N_{0},\ldots ,kN_{0}$; they can be expressed by
(3)$$\left\{\begin{array}{cc} x_{1k}\left(n_{k}\right)=s_{1}\left(n_{k}\right)+\sigma_{1k}\left(n_{k}\right) & \\ x_{lk}\left(n_{k}\right)=\sum_{m=1}^{M}\mu_{lm}s_{1}\left(n_{k}-\Delta \tau_{lm}\right)+\sigma_{lk}\left(n_{k}\right) \end{array}\right.$$

The cross-correlation function of the *k*-th segmented received data and reference data is
(4)$$R_{x_{1k}x_{lk}}\left(\tau \right)=E\left[x_{1k}\left(n_{k}\right)x_{lk}^{H}\left(n_{k}+\tau \right)\right]$$
where E(·) denotes expectation and ${\left(\cdot \right)}^{H}$ represents conjugate transpose. Assuming that the signal and noise are independent, substitute Equation (Equation 3) with Equation (Equation 4), and $R_{x_{1k}x_{lk}}\left(\tau \right)$ can be rewritten as
(5)$$R_{x_{1k}x_{lk}}\left(\tau \right)=\sum_{m=1}^{M}\mu_{lm}^{H}E\left[s_{1}\left(n_{k}\right)s_{1}^{H}\left(n_{k}-\Delta \tau_{lm}+\tau \right)\right]=\sum_{m=1}^{M}\mu_{lm}^{H}R_{s_{1k}}\left(\tau -\Delta \tau_{lm}\right)$$
where $R_{s_{1k}}\left(\tau \right)$ is the self-correlation functions of $s_{1}\left(n_{k}\right)$. Since $x_{1k}\left(n_{k}\right)=s_{1}\left(n_{k}\right)+\sigma_{1}\left(n_{k}\right)$, we have $G_{s_{1k}}\left(\omega \right)=G_{x_{1k}}\left(\omega \right)-G_{\sigma_{1k}}\left(\omega \right)$, where $G_{s_{1k}}\left(\omega \right)$, $G_{x_{1k}}\left(\omega \right)$ and $G_{\sigma_{1k}}\left(\omega \right)$ are the self-spectra of $s_{1}\left(n_{k}\right)$, $x_{1k}\left(n_{k}\right)$ and $\sigma_{1k}\left(n_{k}\right)$, respectively. Perform discrete Fourier transform (DFT) on $R_{x_{1k}x_{lk}}\left(\tau \right)$ and the cross-spectrum of $x_{1k}\left(n_{k}\right)$ and $x_{lk}\left(n_{k}\right)$ is obtained
(6)$$G_{x_{1k}x_{lk}}\left(\omega \right)=G_{s_{1k}}\left(\omega \right)\sum_{m=1}^{M}\mu_{lm}^{H}e^{-j\omega \Delta \tau_{lm}}=\left[G_{x_{1k}}\left(\omega \right)-G_{\sigma_{1k}}\left(\omega \right)\right]\sum_{m=1}^{M}\mu_{lm}^{H}e^{-j\omega \Delta \tau_{lm}}$$
where $\sum_{m=1}^{M}\mu_{lm}^{H}e^{-j\omega \Delta \tau_{lm}}$ is the factor containing multipath delay information.

### 3.2. A DPD Method Based on Subspace Orthogonality in Cross-Spectra

In Equation (Equation 6), the time difference of multipath signals and reference signal is separated by the acquisition of the cross-spectra between them. Equation (Equation 6) can be rewritten as
(7)$$G_{x_{1k}x_{lk}}\left(\omega \right)=G_{x_{1k}}\left(\omega \right)\sum_{m=1}^{M}\mu_{lm}^{H}e^{-j\omega \Delta \tau_{lm}}-\varepsilon_{lk}\left(\omega \right)$$
where $\varepsilon_{lk}\left(\omega \right)=G_{\sigma_{1k}}\left(\omega \right)\sum_{m=1}^{M}\mu_{lm}^{H}e^{-j\omega \Delta \tau_{lm}}$ can be seen as the noise term that obeys the Gaussian distribution. Since the cross-spectrum model (Equation 7) between segmented signals is obtained, high resolution spectrum estimation can be used to estimate the target position. In order to suppress the influence of the noise term $\varepsilon_{lk}\left(\omega \right)$, $G_{x_{1k}x_{lk}}\left(\omega \right)$ is considered to be sampled uniformly to obtain the effective observation vectors. The effective part of $G_{x_{1k}x_{lk}}\left(\omega \right)$ is sampled in frequency domain: (8)$$y_{lk}\left(n_{d}\right)=\sum_{m=1}^{M}\mu_{lm}^{H}G_{x_{1k}}\left(\omega_{k_{d}+\left(n_{d}-1\right)D}\right)e^{-j\omega \Delta \tau_{lm}}-\varepsilon_{lk}\left(\omega_{k_{d}+\left(n_{d}-1\right)D}\right),n_{d}=1,2,\ldots ,N_{d}$$
where $y_{lk}$ is the observation vector of the *k*-th section of the *l*-th received data, $k_{d}$ is the index of the lower bound angle frequency of the cross-spectrum, which is obtained by observation and threshold detection of the cross-spectrum, D(D>M) is the data selection interval, and $N_{d}$ is the dimension of the observation vector. Rewrite Equation (Equation 8) in vector form: (9)$$y_{lk}=A_{lk}\mu_{l}+\varepsilon_{lk}\in C^{N_{d}\times 1}$$
where
(10)$$A_{lk}=\left[a_{lk1},a_{lk2},\ldots ,a_{lkM}\right]\in C^{N_{d}\times M}$$
(11)$$a_{lkm}={\left[G_{x_{1k}}\left(\omega_{k_{d}}\right)e^{-j\omega_{k_{d}}\Delta \tau_{lm}},G_{x_{1k}}\left(\omega_{k_{d}+D}\right)e^{-j\omega_{k_{d}+D}\Delta \tau_{lm}},\ldots ,G_{x_{1k}}\left(\omega_{k_{d}+\left(N_{d}-1\right)D}\right)e^{-j\omega_{k_{d}+\left(N_{d}-1\right)D}\Delta \tau_{lm}}\right]}^{T}\in C^{N_{d}\times 1}$$
(12)$$\mu_{l}={\left[\mu_{l1}^{H},\mu_{l2}^{H},\ldots ,\mu_{lM}^{H}\right]}^{T}\in C^{M\times 1}$$
(13)$$\varepsilon_{lk}={\left[{-\varepsilon }_{lk}\left(\omega_{k_{d}}\right){,-\varepsilon }_{lk}\left(\omega_{k_{d}+D}\right){,\ldots ,-\varepsilon }_{lk}\left(\omega_{k_{d}+\left(N_{d}-1\right)D}\right)\right]}^{T}\in C^{N_{d}\times 1}$$
where ${\left(\cdot \right)}^{T}$ represents transpose. Therefore, the covariance matrix of $y_{lk}$ in Equation (Equation 9) is
(14)$$R_{lk}=E\left[y_{lk}y_{lk}^{H}\right]=E\left[\left(A_{lk}\mu_{l}+\varepsilon_{lk}\right)\left(\mu_{l}^{H}A_{lk}^{H}+\varepsilon_{lk}^{H}\right)\right]=A_{lk}E\left[\mu_{l}\mu_{l}^{H}\right]A_{lk}^{H}+E\left[\varepsilon_{\mathrm{lk}}\varepsilon_{\mathrm{lk}}^{H}\right]=A_{lk}P_{l}A_{lk}^{H}+\sigma_{lk}^{2}I$$
where $R_{lk}\in C^{N_{d}\times N_{d}}$, $P_{l}=E\left[\mu_{l}\mu_{l}^{H}\right]$ and $\sigma_{lk}^{2}I=E\left[\varepsilon_{lk}\varepsilon_{lk}^{H}\right]$. It should be noticed that data segmentation does not change the data correlation, which means the cross-spectrum of each segment of data and reference data should be consistent theoretically. In the same way, the self-spectra of $x_{1k}\left(n_{k}\right)$ should consistent in all the segments. However, the additive noise leads to the difference between the cross-spectra of each segment. In order to balance the influence of noise, replace $G_{x_{1k}}\left(\omega \right)$ with its average $G_{11}\left(\omega \right)$: (15)$$G_{11}\left(\omega \right)=\frac{1}{K}\sum_{k=1}^{K}G_{x_{1k}}\left(\omega \right)$$

Hence, it can be seen that $A_{lk}$ and $a_{lkm}$ are independent of the different segments; that is, the index *k* can be omitted and they can be rewritten as
(16)$$A_{l}=\left[a_{l1},a_{l2},\ldots ,a_{lM}\right]\in C^{N_{d}\times M}$$
(17)$$a_{lm}={\left[G_{11}\left(\omega_{k_{d}}\right)e^{-j\omega_{k_{d}}\Delta \tau_{lm}},G_{11}\left(\omega_{k_{d}+D}\right)e^{-j\omega_{k_{d}+D}\Delta \tau_{lm}},\ldots ,G_{11}\left(\omega_{k_{d}+\left(N_{d}-1\right)D}\right)e^{-j\omega_{k_{d}+\left(N_{d}-1\right)D}\Delta \tau_{lm}}\right]}^{T}\in C^{N_{d}\times 1}$$

Add up the covariance matrices of all the segments and obtain the average covariance matrix: (18)$$R_{l}=\frac{1}{K}\sum_{k=1}^{K}R_{lk}=A_{l}P_{l}A_{l}^{H}+\frac{1}{K}\sum_{k=1}^{K}\sigma_{lk}^{2}I\in C^{N_{d}\times N_{d}}$$

When $P_{l}$ is nonsingular, $R_{l}$ after eigen-decomposition can be rewritten as
(19)$$R_{l}=U_{l_{S}}\Lambda_{l_{S}}U_{l_{S}}^{H}+U_{l_{N}}\Lambda_{l_{N}}U_{l_{N}}^{H}$$
where $U_{l_{S}}\in C^{N_{d}\times M}$ is the signal subspace and $U_{l_{N}}\in C^{N_{d}\times \left(N_{d}-M\right)}$ is the noise subspace that are respectively spanned by the eigenvectors corresponding to the *M* largest and $N_{d}-M$ smallest eigenvalues, and $\Lambda_{l_{S}}$ and $\Lambda_{l_{N}}$ are the diagonal matrices composed of the *M* biggest eigenvalues and $N_{d}-M$ smallest eigenvalues. Based on the orthogonal relation in signal and noise subspace, we have $A_{l}^{H}U_{l_{N}}=0$. From Equation (Equation 16), the relation between $a_{lm}$ and $U_{l_{N}}$ is
(20)$$a_{lm}^{H}U_{l_{N}}=0$$

In a real-world location scenario, the existence of noise leads to the approximate orthogonal relation between $a_{lm}$ and $U_{l_{N}}$. Thus, the estimation of the target source can be obtained by searching the point that satisfies the approximate orthogonal relation. The cost function of any point $q_{0}$ in the chosen region can be established
(21)$$Q\left(q_{0}\right)=\sum_{l=2}^{L}\frac{1}{a_{l}^{H}\left(q_{0}\right)U_{l_{N}}U_{l_{N}}^{H}a_{l}\left(q_{0}\right)}$$
where
(22)$$a_{l}\left(q_{0}\right)={\left[G_{11}\left(\omega_{k_{d}}\right)e^{-j\omega_{k_{d}}\Delta \tau_{l}\left(q_{0}\right)},G_{11}\left(\omega_{k_{d}+D}\right)e^{-j\omega_{k_{d}+D}\Delta \tau_{l}\left(q_{0}\right)},\ldots ,G_{11}\left(\omega_{k_{d}+\left(N_{d}-1\right)D}\right)e^{-j\omega_{k_{d}+\left(N_{d}-1\right)D}\Delta \tau_{l}\left(q_{0}\right)}\right]}^{T}\in C^{N_{d}\times 1}$$
(23)$$\Delta \tau_{l}\left(q_{0}\right)=\frac{∥q_{l}-q_{0}∥}{c}-\frac{∥q_{1}-q_{0}∥}{c}$$

In a multipath environment, $P_{l}$ is not necessarily nonsingular; the solution to this problem is to obtain the smoothed covariance matrix. The specific forward spatial smoothing [40] process is shown in Figure 2, where $y_{lkd}\left(d=1,\ldots ,D\right)\in C^{N_{d}\times 1}$ is the *d*-th observation vector of $G_{x_{1k}x_{lk}}\left(\omega \right)$ and its expression is:(24)$$y_{lkd}={\left[G_{x_{1k}x_{lk}}\left(\omega_{k_{d}+d-1}\right),G_{x_{1k}x_{lk}}\left(\omega_{k_{d}+d-1+D}\right),\ldots ,G_{x_{1k}x_{lk}}\left(\omega_{k_{d}+d-1+\left(N_{d}-1\right)D}\right)\right]}^{T}$$

The smoothed covariance matrix is expressed as
(25)$${\tilde{R}}_{l}=\frac{1}{K}\sum_{k=1}^{K}Y_{lk}Y_{lk}^{H}={\tilde{U}}_{l_{S}}{\tilde{\Lambda }}_{l_{S}}{\tilde{U}}_{l_{S}}^{H}+{\tilde{U}}_{l_{N}}{\tilde{\Lambda }}_{l_{N}}{\tilde{U}}_{l_{N}}^{H}\in C^{N_{d}\times N_{d}}$$
where $Y_{lk}=\left[y_{lk1},y_{lk2},\ldots ,y_{lkD}\right]\in C^{N_{d}\times D}$. Replace $R_{l}$, $U_{l_{S}}$, $U_{l_{N}}$, $\Lambda_{l_{S}}$, $\Lambda_{l_{N}}$ from Equations (Equation 19)–(Equation 21) with ${\tilde{R}}_{l}$, ${\tilde{U}}_{l_{S}}$, ${\tilde{U}}_{l_{N}}$, ${\tilde{\Lambda }}_{l_{S}}$, ${\tilde{\Lambda }}_{l_{N}}$, which are the corresponding results after spatial smoothing. The coordinate of the target source can be obtained by searching the spectral peak of the values of the cost functions over the meshed region.

### 3.3. Detailed Procedures

The main steps of the proposed method are as follows:Data pre-processing: The reference signal $x_{1}\left(n\right)$ is selected and the received long data are segmented as in Equation (Equation 3);Acquisition of cross-spectra: The cross-correlation $R_{x_{1}x_{l}}\left(\tau \right)$ between *l*-th segmented received data and the reference data is calculated in Equation (Equation 5); then, the cross-spectrum $G_{x_{1}x_{l}}\left(\omega \right)$ is obtained by performing DFT to $R_{x_{1}x_{l}}\left(\tau \right)$. In order to reduce the influence of interference, the effective parts in the cross-spectra are selected in Equation (Equation 8). In a multipath environment, spatial smoothing is used to avoid the data coherence in Equation (Equation 24);Construction of cost function and estimation of the source position: The noise subspace is obtained by eigen-decomposition to the screened cross-spectrum. Choose the area of positioning and build the cost function in Equation (Equation 20) at each grid point based on the orthogonal relationship between noise and signal subspace. Then the estimation of the target source is obtained by searching the spectral peak of values of the cost functions at all points.

## 4. Performance Analysis

### 4.1. Complexity Analysis

In this section, the complexity of the proposed DPD method is compared with those of the TDOA method based on generalized cross-correlation (denoted by TDOA-GCC-SCOT) in [12], the TDOA method based on cross-spectrum (denoted by TDOA-MUSIC-NCS) in [18] and the determinant-based DPD method (denoted by DPD-MUSIC-DET) in [33]. The complexities of different methods are listed in Table 1, where *T* is the one-dimensional search times. Specifically, in the proposed and TDOA-MUSIC-NCS methods, the process of cross-spectrum acquisition between two signals with length $\left(2N_{0}-1\right)$ requires $O\left(\left(2N_{0}-1\right)log\left(2N_{0}-1\right)\right)$. Element multiplication of two vectors with length $\left(2N_{0}-1\right)$ requires $O\left(2N_{0}-1\right)$. The process of covariance matrix acquisition of a $\left(N_{d}\times D\right)$ matrix requires $O\left(N_{d}^{2}D\right)$. The eigenvalue decomposition to a $\left(N_{d}\times N_{d}\right)$ matrix requires $O\left(N_{d}^{3}\right)$. The complexity of 2-D peak search of the cost function values in the proposed method is $O\left(\left(L-1\right)T^{2}N_{d}\left(2N_{d}-2M+1\right)\right)$, while that of the DPD-MUSIC-DET method requires $O\left(L\left(L-1\right)T^{2}\left(N_{d}^{3}+LN_{d}^{2}+L^{3}\right)\right)$.

In order to compare the computational complexities of the different methods more intuitively, the variation curves of complexities with $N_{d}$ are shown in Figure 3, where L=4, $N_{l}=16,128$, $N_{0}=256$, K=63, T=50, M=2. It can be seen from Figure 3 that the computational complexity of the proposed DPD method has roughly the same order of magnitude as that of two TDOA methods. Compared with the DPD-MUSIC-DET method, the computational complexity of the proposed DPD method is reduced by two orders of magnitude. Therefore, the proposed DPD method provides an appreciable contribution in reducing the computational complexity, especially compared with other same-type DPD methods.

The real-time performance of the proposed method can be verified by the comparison of the operation time. Perform the four methods 10 times and record their operation time in Table 2, where L=4, $N_{l}=12,288$, $N_{0}=256$, K=48, T=100, M=2. Since the proposed method is based on 2-D peak search, it shows poorer real-time performance than the two TDOA methods. However, compared with the DPD-MUSIC-DET method, the proposed DPD method has a significant improvement in real-time performance, which lies in the great reduction of the dimensions of eigenvalue decomposition and peak search. Therefore, the satisfactory real-time performance of the proposed DPD method is confirmed. In this section, it is demonstrated that the proposed DPD method can reduce the complexity effectively, which guarantees its real-time performance.

### 4.2. Simulation Results

In this section, a quadrature amplitude modulation (QAM) signal whose modulation order is 16 (hereinafter referred to as 16QAM) is used to simulate the transmit signal of the radiation source. The sampling frequency is $f_{s}=$ 125 MHz, the signal bandwidth is B= 40 MHz, the data length is $N_{l}=12,288$, the number of DFT points is $N_{0}=256$, and the number of data segments is K=48. The positioning result of the proposed method is shown in Figure 4, where the signal-to-noise ratio is snr=−5 dB, $k_{d}=200$, M=3, $N_{d}=20$, and D=6. The coordinates of the observation stations are $\left(-500,-300\right)$ m, $\left(-500,300\right)$ m, $\left(500,-300\right)$ m and $\left(500,300\right)$ m. The estimation of the source position is $\left(26,45\right)$ m, while the real position is $\left(26.5,41.5\right)$ m. This simulation testifies to the effectiveness of the proposed DPD method.

The root mean square error (RMSE) is used to evaluate the performance of the proposed method. The RMSE of the estimated position denoted by rmse is given by
(26)$$rmse=\sqrt{\frac{1}{N_{t}}\sum_{i=1}^{N_{t}}∥{{\hat{p}}_{i}-p∥}^{2}}$$
where $N_{t}$ are the simulation times under the same SNR, ${\hat{p}}_{i}$ is the *i*-th estimation to the radiation source, and p is the real position of the source. The RMSE curves of the different methods in a multipath-free environment are shown in Figure 5, where L=4, $f_{s}=$ 125 MHz, $N_{l}=$ 12,288, $N_{0}=256$, K=48, $N_{d}=20$, D=10, T=100, and M=1. Perform $N_{t}=200$ simulations with SNR varying from −10 dB to 15 dB. The RMSE curves versus SNR in a multipath environment are shown in Figure 6, where M=3.

From Figure 5, it is exhibited that the DPD methods performs better than the TDOA methods for the reason of the limitation of the sampling rate in TDE estimation. It can also be seen that the proposed method shows the highest positioning accuracy among these methods. In Figure 6, the RMSE curve of the DPD-MUSIC-DET method becomes higher than that of the TDOA-GCC-NCS method when snr≥ 0 dB. It is evident that the positioning performance of DPD-MUSIC-DET decreases because of the existence of the multipath components. However, the proposed method is not affected by the multipath signals. By comparing Figure 5 with Figure 6, the trend of the RMSE curve of the proposed method remains almost unchanged, which testifies to its effectiveness in resistance to multipath environments.

In order to analyze the relationship between the number of the multipath components and the positioning performance, the RMSE curves versus the number of the multipath components, i.e., *M*, are plotted in Figure 7, where snr=−3 dB. The number of multipath components is increased from M=1 to M=5. At this SNR, the traditional TDOA method cannot estimate the position of the radiation source. The RMSE curves of the four methods rise with the increase of *M*. It is worth mentioning that the TDOA-MUSIC-NCS method performs better than the DPD-MUSIC-DET method when 1<M≤4, which means that the TDOA-MUSIC-NCS has a limited effect against multipath environments. From Figure 7, it can be seen that the RMSE curve versus *M* of the proposed method is the lowest and is hardly rising, which indicates its applicability to multipath environments.

Through a comprehensive comparison of Figure 5, Figure 6 and Figure 7, it can be found that multipath components have little effect on the proposed method, while the other methods are affected to varying degrees. Therefore, the positioning performance of the proposed method in accuracy is considered better than the other methods both in multipath and multipath-free environments.

### 4.3. Real-World Test Results

This section verifies the performance of the proposed method in real-world positioning scenarios, and the schematic diagram of the real-world positioning scenario is shown in Figure 8. The campus positioning scenario is basically composed of four sensors and a radiation source, and the physical drawings of them are shown in Figure 8a,b. Each sensor is equipped with an omnidirectional antenna whose operating frequency band covers 80–8000 MHz and whose power capacity is 25 W. Set Sensor 1 as the reference station, and convert the longitude and latitude coordinates of the four sensors and the radiation source into Cartesian coordinates. The coordinates of the four sensors and source are $\left(0,0\right)$ m, $\left(23.5975,-95.4008\right)$ m, $\left(-60.1265,32.0600\right)$ m, $\left(5.3802,-188.3525\right)$ m, and $\left(-94.5789,-97.4045\right)$ m, respectively, after the conversion. The source transmits the signal to the sensors and the received signals of the sensors are processed in a central site. The modulation mode of the transmit signal is 16QAM and the bandwidth and central frequency are 20 MHz and 720 MHz. In the measured scenario, the overall site is relatively open and there are few multipaths. Figure 8c exhibits the heatmap of the positioning result of the proposed method, where L=4, $f_{s}=125$ MHz, $N_{l}=$ 30,720, $N_{0}=2048$, K=15, D=20, T=400, and $N_{t}=20$. The parameters $N_{d}$ and $k_{d}$ about the effective part adaptively change with the cross-spectra by setting the threshold value. As is shown in Figure 8c, the blue-shaded square represents the region of search, and the red-shaded part is the estimation of the radiation source position, which is close to the real position.

In order to further reflect the superiority of the proposed method, the error cumulative distribution function (CDF) curves are compared among different methods. The error CDF represents the probability of being less than the error value. According to the definition of a CDF curve, it can be concluded that the closer the curve is to the longitudinal axis, the smaller the overall error is. As exhibited in Figure 9, the overall trend of CDF curves is consistent with the RMSE curves in the simulation tests. The error CDF of the proposed method is closest to the longitudinal axis, and nearly 85 percent of the error values are smaller than 10 m. Meanwhile, the other three methods have large error values that are more than 20 m, meaning the failure of the positioning. In fact, the number of multipath components in the real-world test scenario is very few, but the environment noise is more complex. Thus, the DPD-MUSIC-DET method performs better than the TDOA-MUSIC-NCS method in terms of robustness in low-SNR environments. The RMSE of the real-world test is shown in Figure 10. It is clear that the error caused by the proposed method is the smallest in the real-world test. In this section, the superiority in estimation accuracy and availability in practice of the proposed method are proven.

## 5. Conclusions

This paper proposes a TDOA-based DPD method based on the subspace orthogonality of cross-spectra in multipath environments. The data model under multipath environments is built and the method of data processing is introduced. The cost function based on the subspace orthogonal relationship of each cross-spectrum between signals is constructed, and the estimation to the radiation source is obtained by grid search. The computational complexity of the proposed method is proven to be much lower than DPD-MUSIC-DET and slightly higher than TDOA-GCC-SCOT and TDOA-MUSIC-NCS by numerical analysis. Further, RMSE and CDF are used as evaluation indices to testify to the estimation performance in accuracy by simulation test and real-world test. The test results demonstrate the superiority of the proposed method in both multipath and multipath-free environments. The presentation of the TDOA-based DPD method is helpful to realize high accuracy real-time positioning in practice.

## Figures and Tables

**Figure 1 sensors-22-07245-f001:**
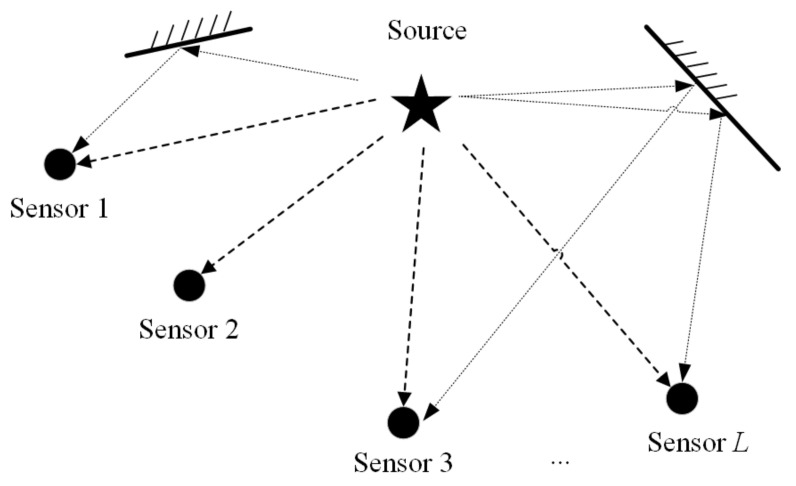
Diagram of a TDOA-based location scenario in a multipath environment.

**Figure 2 sensors-22-07245-f002:**
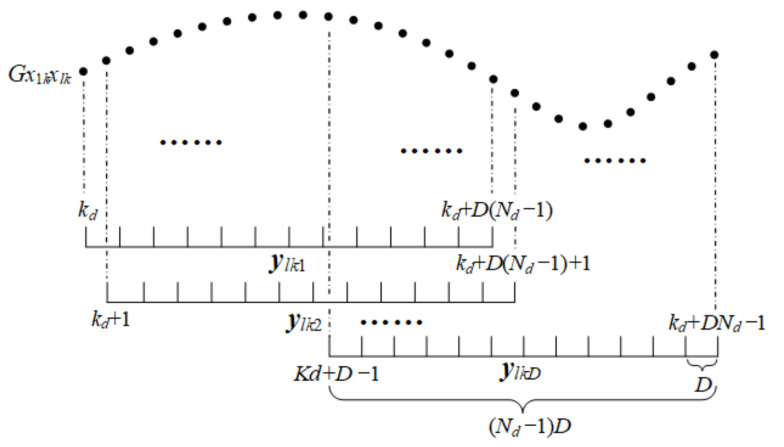
Schematic diagram of the forward spatial smoothing process.

**Figure 3 sensors-22-07245-f003:**
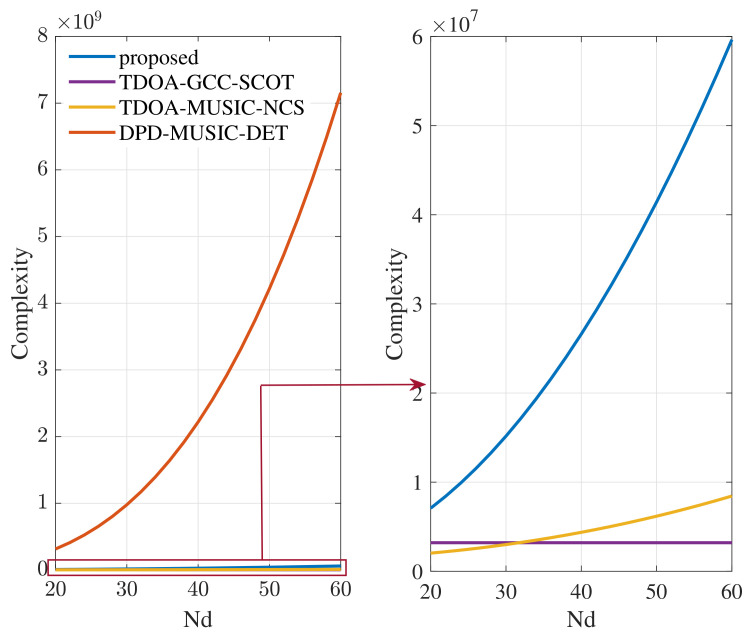
Variation curves of complexities of different methods with $N_{d}$.

**Figure 4 sensors-22-07245-f004:**
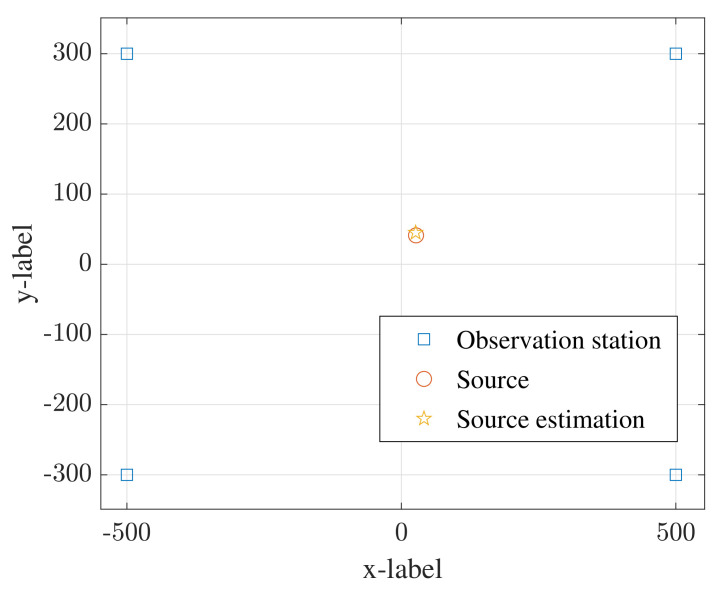
The positioning result of the proposed method $\left(M=3\right)$.

**Figure 5 sensors-22-07245-f005:**
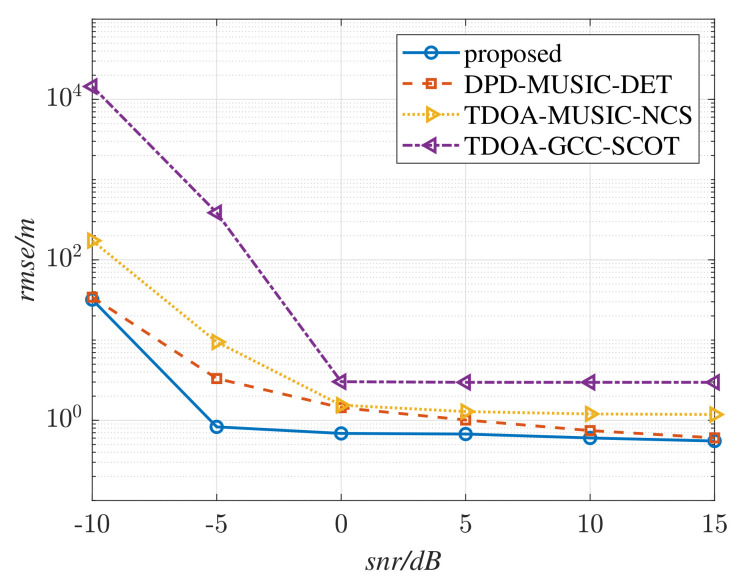
RMSE curves versus SNR in a simulated multipath-free environment $\left(M=1\right)$.

**Figure 6 sensors-22-07245-f006:**
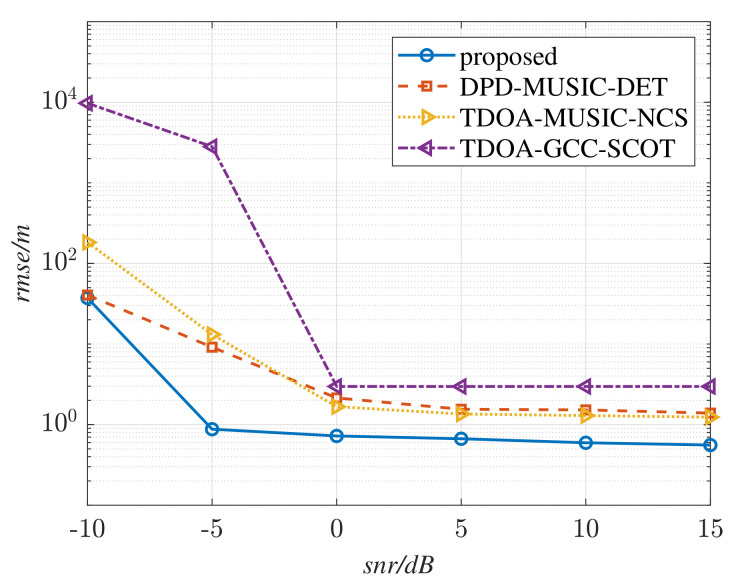
RMSE curves versus SNR in a simulated multipath environment $\left(M=3\right)$.

**Figure 7 sensors-22-07245-f007:**
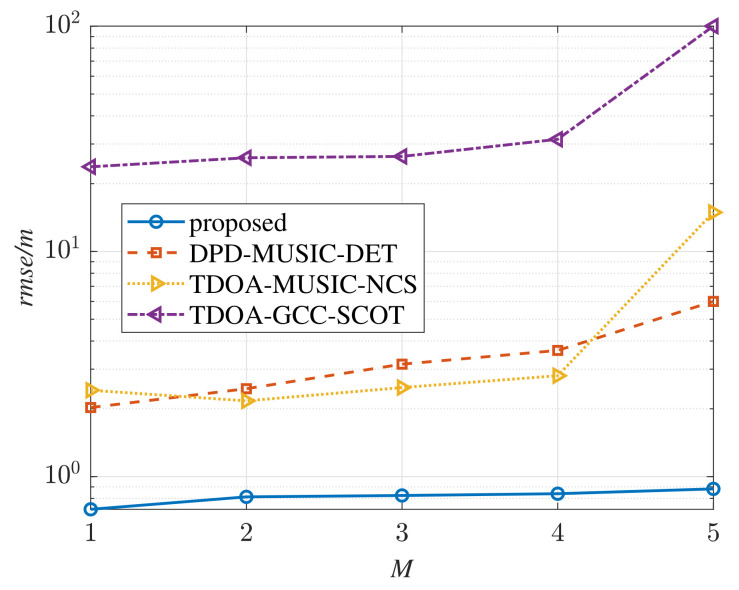
RMSE curves versus the number of multipath components (snr=−3 dB).

**Figure 8 sensors-22-07245-f008:**
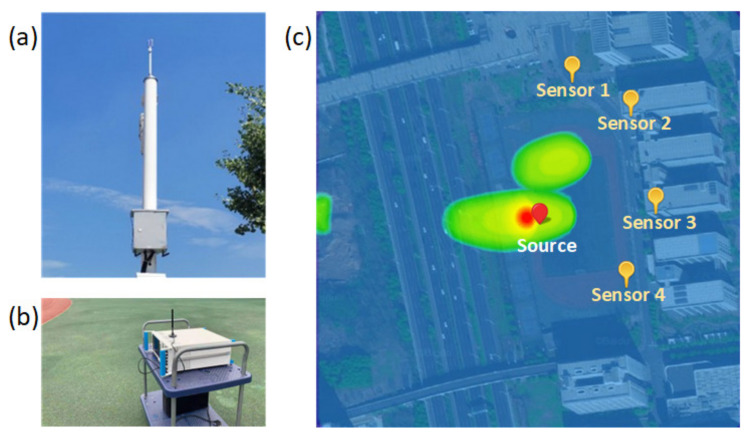
The real-world positioning scenario. (**a**) Physical drawing of a sensor. (**b**) Physical drawing of the radiation source. (**c**) Heatmap of the positioning result by the proposed method.

**Figure 9 sensors-22-07245-f009:**
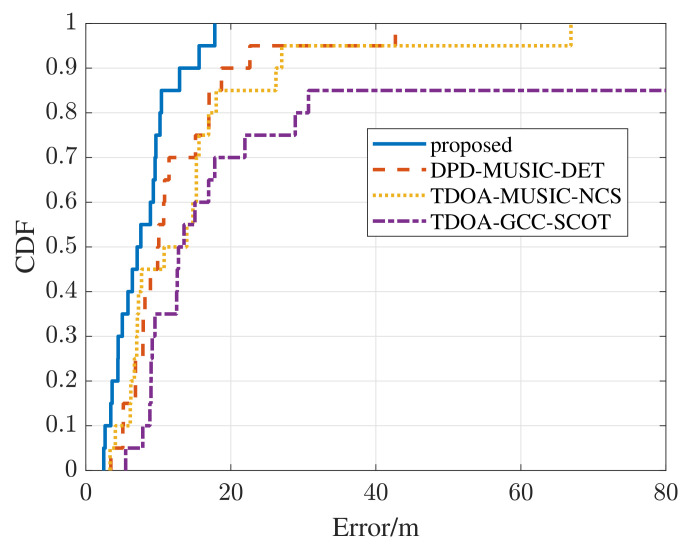
The comparison of error CDF curves in a real-world positioning scenario.

**Figure 10 sensors-22-07245-f010:**
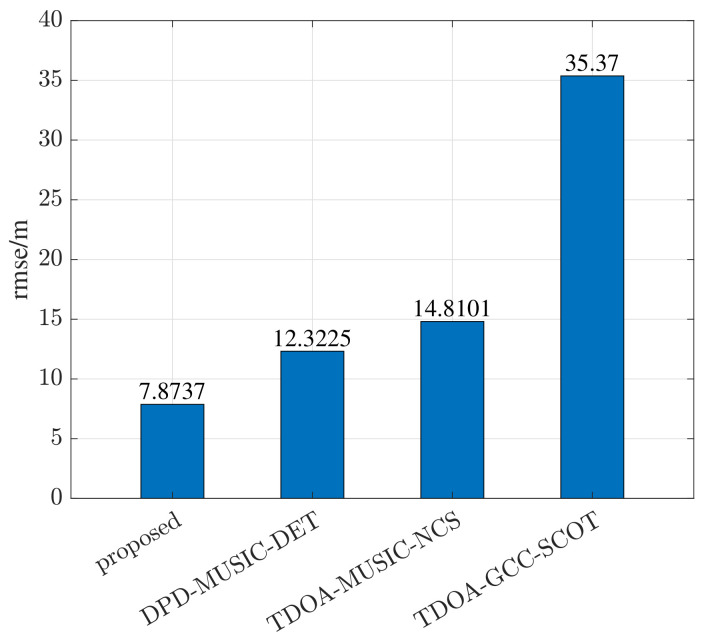
The comparison of RMSE in a real-world positioning scenario.

**Table 1 sensors-22-07245-t001:** Complexities of different methods.

Method	Complexity
Proposed	$O\left(KL\left(2N_{0}-1\right)log\left({2N}_{0}-1\right)+K\left(L-1\right)\left(N_{d}^{2}D+2N_{0}-1\right)\right)+\left(L-1\right)\left(N_{d}^{3}+T^{2}N_{d}\left(2N_{d}-2M+1\right)\right)$
DPD-MUSIC-DET	$O(LKN_{0}logN_{0}+K^{2}N_{d}L^{2}+N_{d}^{3}L^{3}+\left(LN_{d}+\left(L^{2}-L\right)\left(N_{d}^{3}+LN_{d}^{2}+L^{3}\right)T^{2}\right)$
TDOA-MUSIC-NCS	$O(\left(L-1\right)\left(K\left(2\left(2N_{0}-1\right)log\left({2N}_{0}-1\right)+2N_{0}-1+N_{d}^{2}D\right)+N_{d}^{3}+TN_{d}\left(2\left(N_{d}-M\right)+1\right)\right)$
TDOA-GCC-SCOT	$O(\left(L-1\right)\left(3\left({2N}_{l}-1\right)\log\left(2N_{l}-1\right)+{2(2N}_{l}-1\right)))$

**Table 2 sensors-22-07245-t002:** Comparison of operation time.

Method	Proposed	DPD-MUSIC-DET	TDOA-MUSIC-NCS	TDOA-GCC-SCOT
Operation time (s)	1.585086	162.280677	0.371173	0.005917

## Data Availability

Not applicable.

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
