# Peer review of "A Direct Position Determination Method Based on Subspace Orthogonality in Cross-Spectra under Multipath Environments"

_sensors, 2022, doi:10.3390/s22197245_

Round 1

Reviewer 1 Report

The authors claim that they have reduced the computational complexity of the Time-Difference-of-Arrival-based (TDOA) Direct Position Determination (DPD) method ). In addition, they employed the method of multipath scenario. They claim that no recent work is done to use DPD for multipath scenarios. Therefore the novelty is moderate as existing methods are improved. The authors are advised to elaborate on the novelty in the revised manuscript.

Writing is fine but in some places, English grammar requires some. Therefore, for improvement in writing the paper should be reviewed carefully.  

Results are given but few comments.

Although DPD-based methods are not employed for the Multipath scenario there is rich work done for positioning that used methods other than DPD,  the authors are advised to compare the method with some such methods from very recent work.  

Position estimation is normally assessed with a Minimum mean square error between the actual position and estimated position.  Why not here?

Reviewer 2 Report

The paper is well written and proposes an interesting technique. The procedure is well described and demonstrated with experiments.

Some remarks and integrations:

 1-      MUSIC (and its variations)  represents an extremely relevant  reference technique:

https://doi.org/10.1109/TAP.1986.1143830

https://doi.org/ 10.1109/TAP.2012.2232893

2-      The scenario of the proposed methods in literature reported in the introduction refers to rather old papers and does not consider some recent full-hardware solutions described in

https://doi.org/10.1109/TCSII.2020.2995064

https://doi.org/10.1109/TCSI.2020.2979347

3-      The performance analysis proposed in Sec.4  can be improved considering the MUSIC algorithm, at least (a huge literature results are available)

4-      In the proposed experiments it is rather unclear how the Nd, kd and D values are evaluated. Please furnish some more insight on their determination process. In general, a more detailed description of the experimental field could furnish valuable information: distance and position of the considered sources, their power levels, antennas characteristics, and so on. 

5-      Fig.8 compares the CDF for the listed methods but doesn’t give any pratical hint about the expected relative precision/accuracy

Reviewer 3 Report

The paper proposes a "one-step" TDOA-based DPD method based on the subspace orthogonality of cross spectra in environments with (slight) and without multipath.

In contrast to conventional super-resolution methods (such as DPD-DET), a new type of segmentation of the input data saves computing power.

Although the gain in accuracy in (slight) multipath environments (M=3) is shown in the simulation, it leaves questions unanswered:

- Information on the real-time capability of the method is missing.
- The real environment leaves open how strong and what kind of multipath effects are present.
- It is unclear how the number of signals/multipath components "M" is determined: MDL / AIC etc is typically used here?
- Please discuss why in the "multipath" scenario (Fig. 6) a classic error-prone TDOA method (yellow) is just as good as DPD-DET (purple) under multipath, and your method is much better? Whereas in the real world test (Fig. 8), almost all methods are equally good.
It seems as if in the real scenario there is either no multipath and therefore the classic 2-step methods are significantly more robust and accurate or the superresolution methods are worse as the number of signals/components is not approximated well?
- The related work lacks a comparison to the state-of-the-art w.r.t. learning-based methods. I think it's still acceptable if you don't compare yourself to them, but you should at least name and discuss them: supervised methods e.g., direct positioning with CNNs (Niitsoo et al. 2019), ToA parameter estimation for localization with deep learning (Feigl et al. 2021) or unsupervised methods such as channel charting (Studer et al. 2020).

-> Please go into more detail about the conclusions and open questions in the results.

-> Explain the scenarios in more detail.

-> Describe the derivation and approximation of unknown parameters, e.g., M.

-> Assessment of the multipath described in the real test. I'm not clear why TDOA should be worse without multipath. It is known that a proper TOA estimator, a downstream least squares approximate find an optimal position, and a tracking filter on top with a constant velocity model provide the reference (highest accuracy) here, maybe RNNs (see Feigl et al. 2018) outperform them when thy overfit the motion profile. I don't think that a super resolution method can be better here, please discuss, as even "by heart" learning methods are worse here, which have otherwise beaten the state-of-the-art by far in multipath situations.
-> Please discuss those points in detail.
-> M=3 sources are also very few, please discuss, as in typical industrial environments i would rather consider with M=100+. Then, the question of the computing effort in relation to deep learning methods arises.

=> Overall, I like the paper, it is interesting and I would vote  for accepting it, if small things (above) were corrected and spellchecks were carried out.

Reviewer 4 Report

The paper presents an interesting idea that is proven by simulations and live data experiments.
The derivation and description are, however, of very poor quality.
A simple problem is the language, or using mathematical terms - some (not all) examples are:

- "M bigger eigenvalues" -> "M largest eigenvalues"

- "thermodynamic diagram" -> "heatmap"

-  "equation is sampled" (some function is sampled!)

- "covariance matrix of equation" (some random vector has a covariance...)

A more important problem is the lack of a plan for explaining the idea. Authors seem to describe the algorithm steps instead of explaining the method. For example, they introduce segmentation of input data, which complicates the derivation because of an additional index. Later, they talk about smoothing, after which they have to repeat essentially the same equations but with all the variables topped with "~". (eqns 17-22 almost repeated as 23-28, repeating text about eigenvalues, etc.)
I suggest concentrating on the derivation with exact, theoretical values (this may be done in less than one page - if thoroughly planned), then extending the talk about practical aspects.

Minor problems in the idea presentation:

- a value of "D" is used since eqn 8, and the definition of D is not clear

- "The effective part of Equation (7)...." - what does it mean???

- in line 102 authors assume no multipath component in the reference signal - the consequences of the assumption (or its violation) are not discussed,

- eqn 28 we have Q(q_0) on the left side, so it would be welcome to mark which elements of the right side depend on q_0

The complexity analysis shows resulting expressions, but there are no explanations for how they were found.

One sensor has to be chosen as "reference" - how does the choice influence the results?

How was multipath simulated in section 4.2?
